# From complete cross-docking to partners identification and binding sites predictions

**Chloé Dequeker**, **Yasser Mohseni Behbahani**, **Laurent David, Elodie Laine** *,
**Alessandra Carbone** *

Sorbonne Université, CNRS, IBPS, Laboratoire de Biologie Computationnelle et Quantitative (LCQB), Paris, France

* elodie.laine@sorbonne-universite.fr (EL); alessandra.carbone@sorbonne-universite.fr (AC)

## Abstract

Proteins ensure their biological functions by interacting with each other. Hence, characterising protein interactions is fundamental for our understanding of the cellular machinery, and for improving medicine and bioengineering. Over the past years, a large body of experimental data has been accumulated on who interacts with whom and in what manner. However, these data are highly heterogeneous and sometimes contradictory, noisy, and biased. *Ab initio* methods provide a means to a "blind" protein-protein interaction network reconstruction. Here, we report on a molecular cross-docking-based approach for the identification of protein partners. The docking algorithm uses a coarse-grained representation of the protein structures and treats them as rigid bodies. We applied the approach to a few hundred of proteins, in the unbound conformations, and we systematically investigated the influence of several key ingredients, such as the size and quality of the interfaces, and the scoring function. We achieved some significant improvement compared to previous works, and a very high discriminative power on some specific functional classes. We provide a readout of the contributions of shape and physico-chemical complementarity, interface matching, and specificity, in the predictions. In addition, we assessed the ability of the approach to account for protein surface multiple usages, and we compared it with a sequence-based deep learning method. This work may contribute to guiding the exploitation of the large amounts of protein structural models now available toward the discovery of unexpected partners and their complex structure characterisation.

## Author summary

Proteins do not act alone, but perform their biological functions by interacting with each other. However, it is difficult to observe them directly in action, and to collect unbiased clear-cut data on their association. Here, we propose to exploit the protein 3D structures and models accessible nowadays to discover new interactions and alternative binding modes. We simulate the binding of thousands of hundreds protein pairs, and estimate the interaction strength of each pair based on their geometric, physico-chemical and evolutionary properties. We measure proteins' "sociability", and identify a set of putative

**Data Availability Statement:** The data and the analysis scripts are made available at http://www.lcqb.upmc.fr/CCD2PI/.

**Funding:** Funding for open access charge: ANR-18-CE13-0004 (AC); the MAPPING project

(ANR-11-BINF-0003, Excellence Programme "Investissement d'Avenir") (AC); the access to the HPC resources of the Institute for Scientific Computing and Simulation (Equip@Meso project - ANR-10-EQPX- 29-01, Excellence Program "Investissement d'Avenir") (AC, EL). The funders had no role in the study design, data collection and analysis, decision to publish, or preparation of the manuscript.

**Competing interests:** The authors have declared that no competing interests exist.

partners for each protein. We give some guidance for choosing the parameters, and we provide a readout of the predictions. Our approach can complement experimental data, and also predictions produced by machine learning methods relying on protein sequences.

## Introduction

The vast majority of biological processes are ensured and regulated by protein interactions. Hence, the question of who interacts with whom in the cell and in what manner is of paramount importance for our understanding of living organisms, drug development and protein design. While proteins constantly encounter each other in the densely packed cellular environment, they are able to selectively recognise some partners and associate with them to perform specific biological functions. Discriminating between functional and non-functional protein interactions is a very challenging problem. Many factors may reshape protein-protein interaction networks, such as point mutations, alternative splicing events and post-translational modifications [1–5]. Conformational rearrangements occurring upon binding, and the prevalence of intrinsically disordered regions in interfaces further increase the complexity of the problem [6–9]. Ideally, one would like to fully account for this highly variable setting in an accurate and computationally tractable way.

In the past years, a lot of effort has been dedicated to describe the way in which proteins interact and, in particular, to characterise their interfaces. Depending on the type and function of the interaction, these may be evolutionary conserved, display peculiar physico-chemical properties or adopt an archetypal geometry [10–20]. For example, DNA-binding sites are systematically enriched in positively charged residues [10] and antigens are recognized by highly protruding loops [12]. Such properties can be efficiently exploited toward an accurate detection of protein interfaces [10–12, 21–27]. However, the large scale assessment of predicted interfaces is problematic as our knowledge of protein surface usage by multiple partners is still very limited [23].

A related problem is the prediction of the 3D arrangement formed between two or more protein partners. This implies generating a set of candidate complex conformations and correctly ranking them to select those resembling the native structure. Properties reflecting the strength of the association include shape complementarity, electrostatics, desolvation and conformational entropy [28]. Experimental data and evolutionary information (conservation or coevolution signals) may help to improve the selection of candidate conformations [29–31]. To address this problem, molecular docking algorithms have been developed and improved over the past twenty years, stimulated by the CAPRI competition [32–36]. Nevertheless, a number of challenges remain, including the modelling of large conformational rearrangements associated to the binding [32, 37, 38]. Moreover, homology-based modelling often leads to better results than free docking when high-quality experimental data is available.

The development of ultra-fast docking engines exploiting the fast Fourier transform [39–41], deep learning [11] and/or coarse-grained protein models [42] has made large-scale docking computational experiments feasible. Moreover, the availability of 3D structural models from AlphaFold for entire proteomes [43] has dramatically expanded the applicability of docking algorithms. This favourable context renders protein-protein interaction network reconstruction accessible at a very large scale by *ab initio* approaches that avoid biases coming from experimental conditions and allow for a blind search for partners that may lead to the discovery of new interactions.

In a large-scale docking experiment, hundreds or thousands of proteins are either docked to each other (complete cross-docking, CC-D) or to some arbitrarily chosen proteins. The generated data can be straightforwardly exploited to predict protein interfaces [23, 44–47]. Indeed, randomly chosen proteins tend to dock to localised preferred regions at protein surfaces [48]. In this respect, the information gathered in the docking experiment can complement sequence- and structure-based signals detected within monomeric protein surfaces [23]. Beyond interface and 3D structure prediction, very few studies have addressed the question of partner identification. The latter has traditionally been regarded as beyond the scope of docking approaches. However, an early low-resolution docking experiment highlighted notable differences between interacting and non-interacting proteins [49], and we and others [50–53] have shown that it is possible to discriminate cognate partners from non-interactors through large-scale CC-D experiments. An important finding of these studies, already stated in an earlier experiment involving 12 proteins [54], is that relying on the energy function of the docking algorithm is not sufficient to reach high accuracy. This holds true for shape complementarity-based energy functions [50], and also for those based on a physical account of interacting forces [53, 54]. Nevertheless, combining the docking energy with a score reflecting how well the docked interfaces match experimentally known interfaces allows reaching a very high discriminative power [53]. Moreover, the knowledge of the global social behaviour of a protein can help to single out its cognate partner [50, 53]. That is, by accounting for the fact that two proteins are more or less *sociable*, we can lower down or lift up their interaction strength, and this procedure tends to unveil the true interacting partners [50]. This notion of sociability also proved useful to reveal evolutionary constraints exerted on proteins coming from the same functional class, toward avoiding non-functional interactions [50].

In principle, the estimation of systemic properties such as residue binding propensity and protein sociability shall be more accurate as more proteins are considered in the experiment. But the problem of discriminating them will also become harder. When dealing with several hundreds of proteins, the correct identification of the cognate partners requires an incredible accuracy as they represent only a small fraction of the possible solutions. For instance, a set of 200 proteins for which 100 binary interaction pairs are known will lead to the evaluation of 40 000 possible pairs, and for each pair several hundreds of thousands candidate conformations (at least) will have to be generated and ranked.

Here, we present a general approach for the identification of protein partners and their discrimination from non-interactors based on molecular docking. Like our previous efforts [50, 53, 54], this work aims at handling large ensembles of proteins with very different functional activities and cellular localisations. Although these classes of proteins appear to have different behaviours, we approach the problem of partner identification from a global perspective. We report on the analysis of data generated by CC-D simulations of hundreds of proteins. We combine together physics-based energy, interface matching and protein sociability, three ingredients we previously showed to be relevant to partner identification and discrimination [50, 53, 54]. We move forward by investigating what other types of information may be needed to improve the discrimination. To this end, we systematically explore the space of parameters contributing to partner identification. These include the scoring function(s) used to evaluate the docking conformations, the strategy used to predict interacting patches and the size of the docked interfaces. We show that our approach, CCD2PI (for "CC-D to Partner Identification"), reaches a significantly higher discriminative power compared to a previous study addressing the same problem [53]. We demonstrate that this result holds true overall and also for individual protein functional classes. Our results emphasise the importance of the docking-inferred residue binding propensities to drive interface prediction, and the positive contribution of a statistical pair potential to filter docking conformations. We define a set of default

parameter values, with minimal variations between the different classes, for practical application to any set of proteins. Importantly, we place ourselves in a context where we do not know the experimental interfaces and use predictions instead. To evaluate CCD2PI predictions, we consider structurally characterised interactions coming from the Protein Data Bank (PDB) [55] as our gold standard. We primarily consider the docking benchmark annotations [56], and we extend them by transferring knowledge from complex structures involving the same or very similar proteins. This strategy is supported by the observation that functional interfaces are conserved across closely related homologs [57]. Moreover, previous works from us and others have emphasised its biological pertinence and usefulness to evaluate protein-protein/DNA/RNA interface prediction methods [23, 58]. We show that the protein interaction strengths computed by CCD2PI are in good agreement with available structural data. We discuss the implications of these strengths for protein functions. This work paves the way to the automated *ab initio* reconstruction of protein-protein interaction networks with structural information at the residue resolution. Since, the reconstruction is based on docking calculations, it not biased by specific targets nor by the limitations of experimental techniques.

## Results

### Computational framework

The workflow of CCD2PI is depicted in Fig 1. We exploit data generated by CC-D experiments performed on hundreds of proteins. In the present work, the CC-D was performed using the rigid-body docking tool MAXDo [54]. The proteins are represented by a coarse-grained model and the interactions between pseudo-atoms are evaluated using Lennard-Jones and Coulombic terms [42]. For each protein pair, MAXDo generated several hundreds of thousands of candidate complex conformations (Fig 1, top left panel). Each one of these conformations is evaluated by computing the product of the overlap between the docked interface (DI) and some reference interface (RI), a docking energy (either from MAXDo or another one, see Materials and methods), and a statistical pair potential [59] (optional). By formulating the score as a product, we effectively use the interface overlap, the docking energy and the pair potential as successive filters to select the best conformation. The rationale is that ideally, the selected conformation should meet all three criteria: match the expected interface, be energetically favourable, and reflect the amino-acid pairing preferences found in experimental complexes. For instance, let us consider a conformation displaying a perfect interface overlap, but with the interacting surface of the ligand rotated by 180˚ with respect to that of the receptor. It would have a very low fraction of native contacts, and we expect it to be correctly filtered out by the docking energy and/or the pair potential. We detect the DIs based on interatomic distances using our efficient algorithm INTBuilder [60]. To place ourself in a realistic scenario, we predict the RIs, instead of extracting them from the known complex structures. Our predictive algorithm relies on sequence- and structure-based properties of single proteins [12], as well as a systemic property, namely residue binding propensities inferred from the CC-D [23] (see Materials and methods). Formally, given two proteins $P_1$ and $P_2$, we estimate the interaction index of $P_1$ with respect to $P_2$ as

$$II_{P_1,P_2} = min(FIR_{P_1,P_2} \times E_{P_1,P_2}[\times PP_{P_1,P_2}]), \tag{1}$$

where $FIR_{P_1,P_2}$ (Fraction of Interface Residues) is the fraction of the DIs composed of residues belonging to the (predicted) RIs for the two proteins, $E_{P_1,P_2}$ is the docking energy (negative value) and $PP_{P_1,P_2}$ is a pair potential score which may or may not be included in the formula. The latter evaluates the likelihood of the observed residue-residue interactions and might

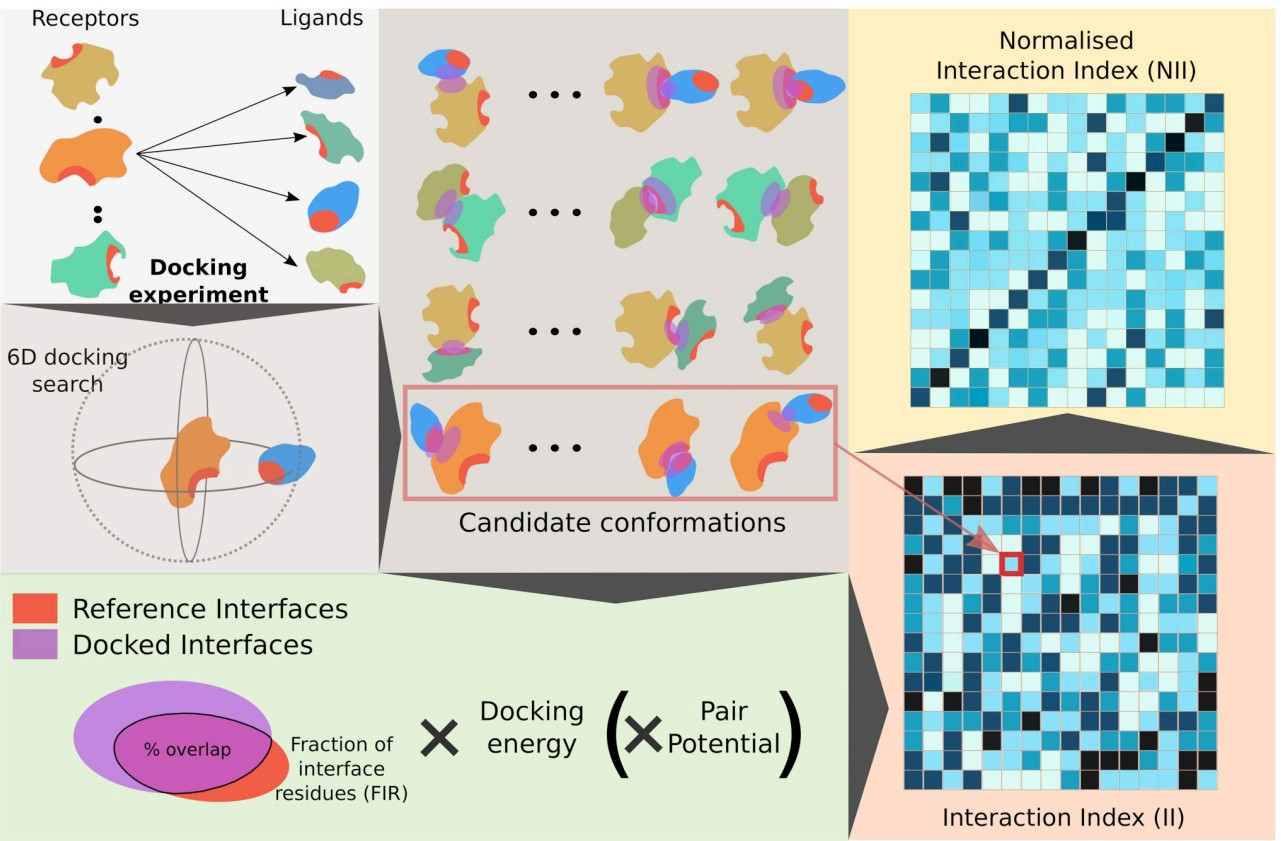

**Fig 1. Principle of the method.** We start from an all-to-all docking experiment (top left panel). Each protein is docked to all proteins in the set. By convention, in each docking calculation, we define a *receptor* and a *ligand*. The red patches on the protein surfaces correspond to predicted interfaces. For a given protein pair $P_1 P_2$, we generate a pool of conformations associated with energies (top middle panel). Here, both the predicted interfaces and the docked interfaces are highlighted by patches, in red and purple respectively. One can readily see whether they overlap or not. The extent of this overlap (Fraction of Interface Residue) is multiplied by the docking energy to evaluate each docking conformation (bottom left panel). Optionally, we also consider a statistical pair potential in the formula. The best score is computed over all docking conformations and assigned to the protein pair. By doing the same operation for all pairs we compute a matrix of interaction indices (bottom right panel, the darker the higher). If the receptor and the ligand play equivalent roles in the docking calculations, then the matrix will be symmetrical. Otherwise, two different docking calculations are performed for each protein pair $P_1 P_2$ and the matrix will be asymmetrical, as shown here. These indices are then normalised to account for proteins' global social behaviour, hopefully allowing for singling out the cognate partners (top right panel). In the example here, the cognate pairs are ordered on the diagonal.

bring complementary information with respect to the docking energy. We use CIPS [59], a high-throughput software designed to swiftly reduce the search space of possible native conformations with a high precision. The minimum is computed over the whole set or a pre-filtered subset of docking conformations (see Materials and methods). One should note that in the general case, $II_{P_1,P_2}$ and $II_{P_2,P_1}$ come from two different docking runs and are not necessarily equal. This is because the receptor and ligand surfaces are not explored in an equivalent manner by the docking algorithm (see Materials and methods).

The computed interaction indices (Fig 1, matrix at the bottom right) are then normalised to account for the protein global social behaviour. Formally, the *II* values are weighted using the sociability index (S-index) [50], defined as

$$S_{P_i} := \frac{1}{2|\mathcal{P}|} \sum_{P_j \in \mathcal{P}} II_{P_i,P_j} + II_{P_j,P_i}, \qquad (2)$$

where $\mathcal{P}$ is the ensemble of proteins, including $P_i$. The normalised interaction index *NII* between $P_1$ and $P_2$ is computed as a symmetrised ratio of interaction indices (see Materials and methods). Finally, the *NII* values are scaled between 0 and 1 and $NII_{P_1,P_2} = 1$ when $P_2$ is the protein predicted as interacting the most strongly with $P_1$ (Fig 1, matrix on the top right).

## CCD2PI accurately singles out cognate partners within specific functional classes

We assessed the discriminative power of CCD2PI on a set of 168 proteins forming 84 experimentally determined binary complexes (Protein-Protein Docking Benchmark v2, PPDBv2, see Methods). Here, we place ourselves in a context where we seek to identify one "true" partner, annotated in the PPDBv2, for each protein from the benchmark. Over all possible 28 224 interacting pairs, the cognate partners were singled out with an Area Under the Curve (AUC) of 0.67 (Fig 2A). On average, 3 putative partners were predicted with a *NII* score above 0.8, and about 10 above 0.6, for each given protein (Fig 2C and S1 Fig). Hence, CCD2PI assigns high interaction strengths to a relatively small number of pairs, compared to the enormous number of potential pairs. In this respect, the contribution of the normalisation stands out as instrumental (S2A and S2B Fig, compare the number of dark spots between the *II* and *NII* matrices). By lowering down the interaction strengths computed for highly sociable proteins, it eliminates most of the "incorrect" partners. Given a protein, only the putative partners binding favourably to it, with a high *II* score, and in a specific manner, as indicated by a low S-index, stand out after the normalisation. This effect is illustrated by S3 Fig on the human GTPase-activating protein p120GAP and gonadotrophin.

The docking energy and the pair potential in Eq 1 (*II* formula) will favour the protein pairs whose RIs have a high physico-chemical and shape complementarity. Consistently, we observed that the RIs of the proteins predicted as plausible partners for a given protein share some common 3D physico-chemical patterns. For instance, we can clearly identify a pattern of positively charged residues common to the RIs of the "incorrect" top 5 predicted partners for the human GTPase-activating protein p120GAP (1WQ1_l) and the RI of its cognate partner H-RAS, ranked at the 6th position (S3A Fig). In the case of the human gonadotrophin (1QFW_l), the RI of its cognate antibody, ranked 13th, displays an enrichment in negatively charged and aromatic residues, also observed for the RIs of the "incorrect" top 5 predicted partners (S3B Fig).

We further assessed CCD2PI's ability to identify the PPDBv2 cognate partners among proteins coming from the same functional class (Fig 2A, blue curve). The partnerships between bound antibodies and their antigens (*ABA*), between enzymes and their inhibitors, substrates, or regulators (*EI*, *ES*, *ER*) and between the other proteins and their receptors (*OR*) are particularly well detected (AUC>0.75). By contrast, the subset regrouping everything that could not be classified elsewhere (others, *OX*) is the most difficult to deal with. This subset likely contains proteins involved in signalling pathways and establishing transient interactions through modified sites, such as phosphorylated sites. As a consequence, correctly predicting their interfaces may be particularly challenging. Conformational changes occurring upon binding seem to play a role as the antibody-antigen cognate pairs are better detected when the antibodies are bound (Fig 2A, compare *AA* and *ABA*).

The AUC values achieved by CCD2PI are systematically and significantly better than those computed with our previous pipeline (Fig 2A, compare the blue and purple curves), or similar in the case of the other-with-G-protein class (*OG*). Replacing the predicted RIs by the interfaces extracted from the PDB complex structures, which can be seen as *perfect* predictions, leads to increased AUC values for almost all classes (Fig 2A, areas in grey tones, and

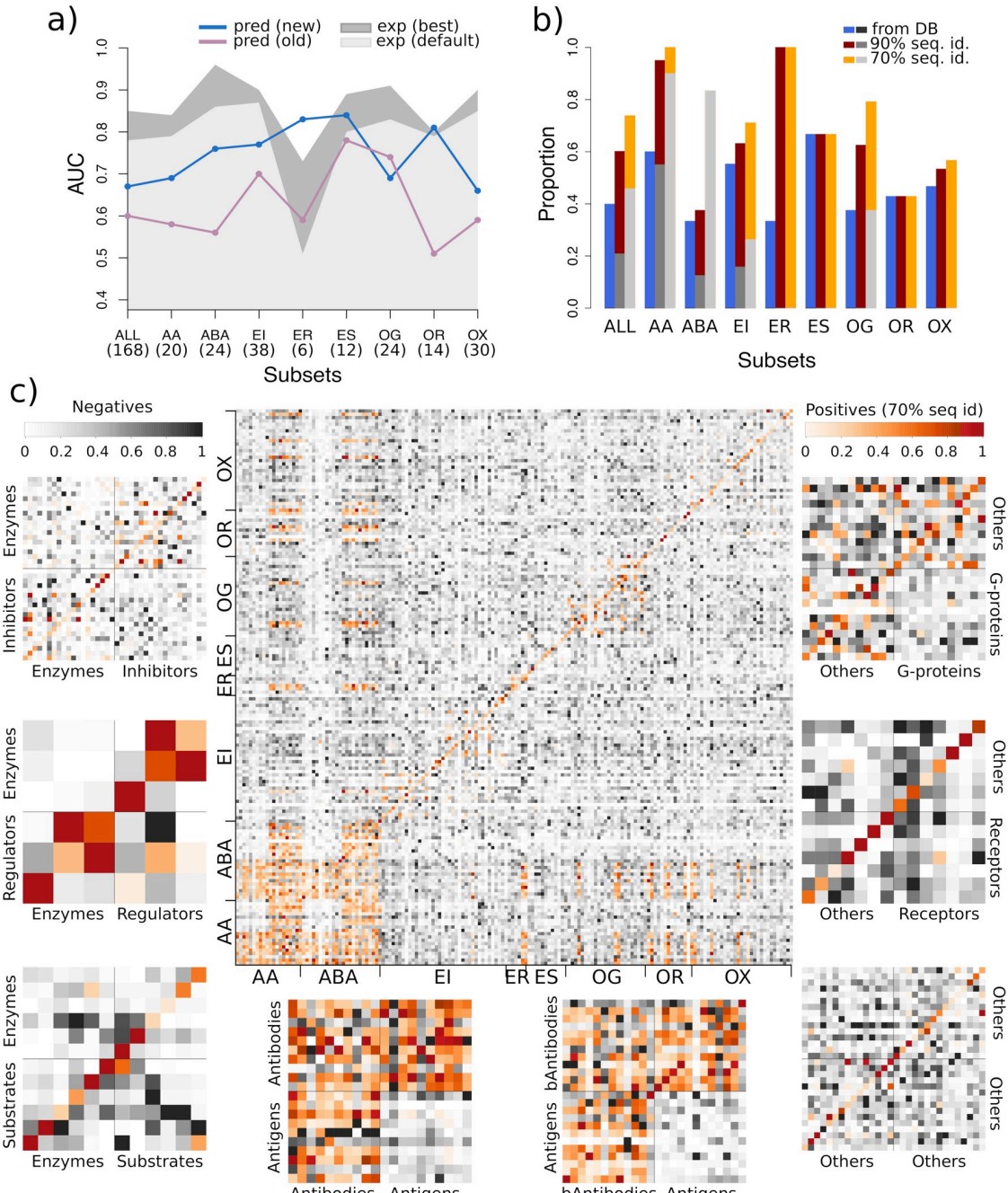

**Fig 2. Predictive performance on the PPDBv2.** (A) AUC values computed for the whole dataset and for the different functional classes. For each protein, we consider one "true" cognate partner, defined from the PPDBv2 annotations. The results obtained with CCD2PI are indicated by the blue curve. For comparison, we also show the results reported in [53] in purple. The areas in grey tones give the discriminative power reached when exploiting the knowledge of the experimental interfaces, using either our default parameters (in light gray) or parameters optimized for such interfaces (in dark grey, see also Materials and methods). The number of proteins in each subset is indicated in parenthesis. (B) Proportion of proteins with at least one known partner found in the top 20% of CCD2PI predictions, for each subset. The known partners are defined from the PPDBv2 annotations (in blue) or are inferred from complex PDB structures involving the proteins from the set or their close homologs, sharing more than 90% (in dark red) or 70% (in orange) sequence identity. The grey bars give baseline expected values based on the number of known partners (see Materials and methods). (C) *NII* matrices computed by CCD2PI. The proteins are ordered on the x-axis such that the *receptors* (*e.g.* antibodies) appear first, and then the *ligands* (*e.g.* antigens). They are ordered on the y-axis such that the cognate pairs annotated in PPDBv2 are located on the diagonal. The orange tones highlight the experimentally known interacting pairs (annotated in the PPDBv2 and transferred by homology). AA: antibody-antigen, ABA: bound antibody-antigen. EI: enzyme-inhibitor. ER: enzyme with regulatory or accessory chain. ES: enzyme-substrate. OG: other-with-G-proteins. OR: other-with-receptor. OX: others.

S2C and S2D Fig). This suggests that proteins competing for the same region at the protein surface do not target exactly the same set of residues. Knowing exactly which residues are involved in an interaction greatly helps in the identification of the partner. Of course, this *perfect* knowledge is generally inaccessible in a fully predictive context. In fact, the predicted interfaces might give a more realistic view on protein surface usage since they tend to better match *interacting regions* [23], defined from several experimental structures and representing the interface variability induced by molecular flexibility and multi-partner binding. Notice-ably, the advantage of experimental over predicted RIs reduces or even cancels out for the small subsets (<15 proteins, *ER*, *ES* and *OR*). This suggests that approximations in the definition of the interfaces do not influence partner identification when few proteins are considered.

## The interaction strengths predicted by CCD2PI reveal the multiplicity of protein interactions

To estimate the agreement between the interaction strengths predicted by CCD2PI and experi-mental data, we extended the set of "true" partners by homology transfer. Specifically, we looked in the PDB for 3D structures of complexes involving the proteins from PPDBv2 or their close homologs (see Materials and methods). We considered that a structurally character-ised interaction found for $P'_1$ and $P'_2$, sharing a high sequence similarity with $P_1$ and $P_2$, respec-tively, was a strong indicator of the possibility for $P_1$ and $P_2$ to interact with each other. We identified 585 interacting pairs from homologs sharing more than 90% sequence identity with the proteins from PPDBv2, and 1 834 at the 70% sequence identity level (Fig 2C, cells colored in orange). These high levels of sequence similarity ensure a high confidence in the newly detected interactions, although homology transfer *per se* does not guarantee they are functional in the cell. We observed the biggest increase in the number of partners for the antibodies (Fig 2C, S4A, S4B and S4C Fig). Some of the homology-transferred partners are direct competitors of the cognate partners annotated in PPDBv2 as they target the same region at the protein sur-face. Depending on the approximations in the predicted RIs, the former may be more favoured than the latter by CCD2PI. A few examples of homology-transferred partners better ranked than the PPDBv2-annotated partners are shown in S5 Fig. Overall, the probability of finding at least one "true" partner in the top 20% predictions is almost systematically increased when extending the set of positives (Fig 2B). For instance, 71% (27 out of 38) of the proteins from the *EI* subset have at least one partner inferred at more than 70% sequence identity ranked in the top 7. Moreover, the homology-transferred interactions tend to populate the regions of the matrices displaying high interaction strengths (Fig 2C and S4D Fig). For instance, CCD2PI predictions suggest that antigens tend to avoid each other much more than antibodies, and indeed much more homology-transferred interactions are found among antibodies, compared to antigens (*AA* and *ABA*). A similar trend is also observed for the enzyme-regulator (*ER*) and enzyme-substrate (*ES*) and other-with-G-protein (*OG*) subsets (Fig 2C and S4D Fig). We observe more predicted and experimental regulator-regulator and substrate-substrate interac-tions than enzyme-enzyme interactions, and more other-other interactions than interactions among G proteins.

## The ingredients of partner discrimination

CCD2PI comprises four main hyper-parameters potentially influencing the results (Table 1), namely (a) the distance threshold used to detect the DIs, (b) the scoring strategy used to pre-dict the RIs, (c) the docking energy function used to compute *II*, and (d) the optional inclusion of the pair potential in the *II* formula. The distance threshold modulates the size of the DIs

**Table 1. Main hyper-parameters of CCD2PI.**

| Docked interfaces Distance threshold (in Å) | Predicted interfaces Scoring strategy | Docking energy[a] ($E$) | Pair potential [b] ($PP$) |
|---|---|---|---|
| 4.5 | SC-mix | **MAXDo** | **CIPS** |
| **5** | SC-monoSeed-mix | iATTRACT | None |
| 6 | **SC-dockSeed-mix** | PISA | |
| | SC-juxt | | |

The default parameter values are highlighted in bold. They were optimized on PPDBv2 (see Methods).

[a] MAXDo was chosen for all functional classes but EI and ER, where it was replaced by PISA and iATTRACT respectively.

[b] CIPS was used for all functional classes but OR.

while the scoring strategy influences how close the RIs are from the experimentally known interfaces. The choice of the energy function and that of using or not the pair potential directly impact the calculation of the interaction index. In order to avoid the risk of overfitting, we strove to determine global default parameter values (Table 1, see also Materials and methods). In the following, we report on a systematic analysis of the influence of the parameters on the discriminative power of the approach, also by considering functional classes (Fig 3). The total number of possible parameter combinations is 72, and we focused on the top 15, for the whole dataset and for its eight subsets. Given a parameter under study, the pool of 15 top combinations was divided by the set of possible values for the parameter (see Materials and methods).

The estimation of the match between the DIs and the RIs depends on the way the former are detected and on the strategy adopted to predict the latter. We observed that varying the distance threshold used to detect the DIs between 4.5 and 6Å does not significantly impact the discrimination on the whole dataset, nor on most of the functional classes (Fig 3A). Nevertheless, it is clearly preferable to define smaller than bigger DIs for the identification of antibody-antigen cognate pairs (Fig 3A, see *AA* and *ABA*). Interestingly, this trend is not observed when using experimental interfaces as RIs (S5B Fig). This suggests that as the DIs grow, residues not specific to the cognate interactions but present in the predicted RIs are being considered. To predict interfaces, we considered four main strategies, each one of them comprising between 3 and 4 scoring schemes (S6 Fig and see Materials and methods). Our algorithm relies on four descriptors, evolutionary conservation, physico-chemical properties, local geometry and docking-inferred binding propensities, and the strategies differ in the way we combine these properties. The one leading to the best results on the whole dataset and also on a couple of functional classes is SC-dockSeed-mix (Fig 3B, see *ABA* and *OX*). In this scoring scheme, the *seed* of the predicted interface is defined based on the propensities of protein surface residues to be targeted in the docking calculations. Then, the seed is extended combining these docking propensities with evolutionary, geometrical and physico-chemical properties (see Materials and methods). The strategy leading to the worst results, SC-monoSeed-mix, introduces the docking propensities only after seed detection. The seeds are detected because they are highly conserved or protruding. SC-monoSeed-mix is not even found in the top 15 combinations of parameters for the whole dataset, nor for the enzyme-substrate and *other* classes (Fig 3B). This analysis emphasises the crucial role of the docking propensities to drive the interface predictions.

Regarding the docking energy, we considered MAXDo, iATTRACT and PISA. MAXDo and iATTRACT are very similar as they include the same contributions (see Materials and methods). They mainly differ in the treatment of the clashes, better tolerated in iATTRACT, and of the electrostatic contribution, more persistent at long distances in iATTRACT. PISA is different as it estimates the likelihood of a macromolecular assembly to be functionally relevant

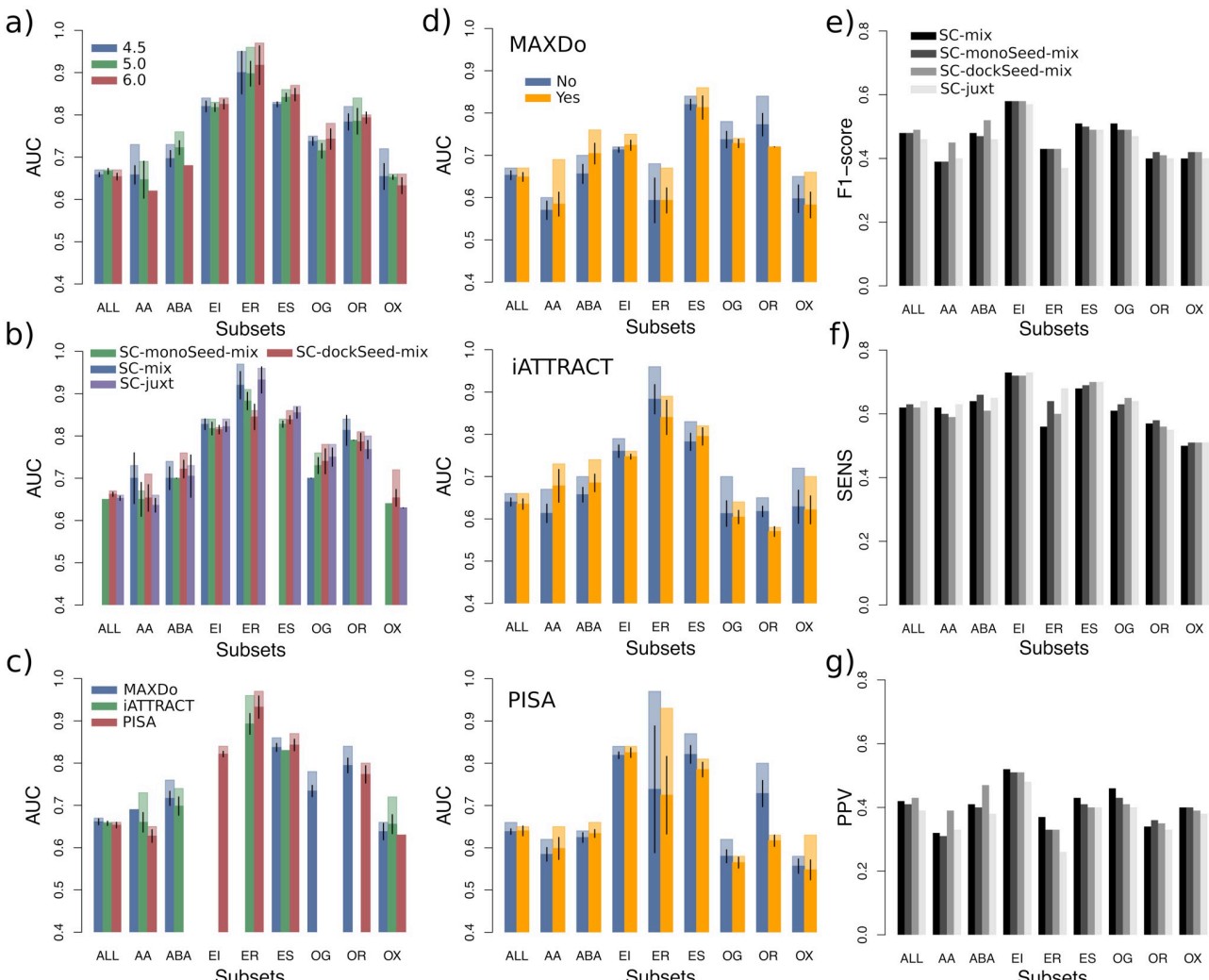

**Fig 3. Influence of the parameters for PPDBv2 when considering predicted RIs. (A-D)** Variation of the AUC values upon parameter changes. The four parameters considered are: **(A)** the distance threshold used to define docked interfaces, **(B)** the scoring strategy used to predict interfaces, **(C)** the docking energy, and **(D)** the presence or absence of the pair potential, depending on the docking energy. In each plot, for each protein class, we considered the 15 combinations yielding the highest AUC values, among all 72 possible combinations. For a given parameter, the different bars correspond to a partition of this combination set according to the possible values of the parameter. If a parameter value was not present in the 15 best combinations, then it does not appear on the plot. We report the average AUC values (in opaque) and the maximum AUC values (in transparent). The black segments indicate the intervals $[\mu - 2\sigma_\mu, \mu + 2\sigma_\mu]$, where $\mu$ is the mean and $\sigma_\mu$ is the standard error of the mean. **(E-G)** Resemblance between predicted and experimental interfaces. **(E)** F1-score. **(F)** Sensitivity. **G)** Positive predictive value.

based on chemical thermodynamics (see Materials and methods). While all three energies perform almost equally well on the whole dataset, with a little advantage for MAXDo, the results on the individual subsets are more contrasted (Fig 3C). In particular, PISA is the only energy function appearing in the top 15 combinations for the enzyme-inhibitor subset (*EI*) while MAXDo is the only one for the other-with-G-protein subset (*OG*). Finally, we investigated the influence of including or not the statistical pair potential CIPS to compute the interaction index (Fig 3D). While CIPS improves the discrimination for the antibody-antigen subsets (*AA* and *ABA*), it is clearly detrimental for the other-with-receptor class (*OR*). The extent of these impacts may vary depending on the energy function with which CIPS is paired, but the trends are consistent from one energy function to another. The picture is very different when we

replace the predicted RIs by experimental interfaces (S7D Fig). In this context, CIPS is mostly contributing in a negative way to the identification of the cognate partners. This suggests that CIPS may underrate some near-native conformations. Although this would not affect much the results when the RIs are predicted, since the number of incorrect conformations removed largely surpasses the number of near-native conformations wrongly removed, this could prove detrimental when using the experimental interfaces, especially in a context where the number of positives is very small compared to that of negatives.

## Small approximations in the reference interfaces may significantly impact partner identification

We further characterised the relationship between the ability of singling out cognate partners and the resemblance between the predicted and the experimental interfaces. The average F1-values of the predicted interfaces range between 0.37 and 0.58 (Fig 3E). The strategy leading to the best AUC values for partner discrimination, namely SC-dockSeed-mix, gives the most accurate predicted interfaces overall (Fig 3E, 3F and 3G, *ALL*). It is also significantly more precise than the other strategies in the detection of the antibody-antigen interfaces (Fig 3E, 3F and 3G, *AA* and *ABA*). Looking across the different classes, it is *a priori* not obvious to assess a direct correlation between the quality of the predicted interfaces and the discriminative power of the approach. In particular, the three subsets (*ER*, *ES* and *OR*) for which predicted RIs lead to AUCs as good as those obtained with experimental RIs (Fig 2A) do not stand out for the quality of their predicted interfaces (Fig 3E, 3F and 3G). This confirms that when dealing with few proteins ($<15$), working with approximate interfaces do not hamper the identification of the cognate partners. However, if we disregard these subsets, then we find that the ability to detect the cognate pairs is highly correlated with the F1-score and the precision of the predicted interfaces (S8 Fig). The Pearson correlation coefficient is of 0.86 (resp. 0.90) between the AUC values and the F1-scores (resp. positive predictive values, PPV) computed for SC-dockSeed-mix. Focusing on the 16 proteins for which the F1-score is very low ($<0.2$), we found that replacing the predicted interfaces by the experimental ones largely improves the ability to single out the cognate partner in half of the cases (S9 Fig). Nevertheless, in the remaining half, improving interface quality brings little gain to partner identification, or even has a deleterious impact. In five cases, the cognate partner is even identified in the top 20% despite the low quality of the predicted RI. These results reveal the existence of protein surface regions onto which cognate partners bind more favourably than non-interactors, although they have not been experimentally characterised as directly involved in the interaction. We hypothesise that these regions might correspond to alternative binding modes with the cognate partner.

To investigate more precisely the sensitivity of partner discrimination with respect to approximations in the RIs, we generated shifted decoys from the experimental interfaces. For each interface in the dataset, we moved between 10 and 100% of its residues, by increments of 10% (see Materials and methods). This allowed us to control the deviation of our RIs with respect to the experimentally known interfaces of the cognate interactions. We observed that the AUC computed for partner identification decreases as the shifted decoys share less and less residues in common with the experimental interfaces (Fig 4). The only notable exception is the smallest class, namely *ER*, which displays a chaotic behaviour. The two other smallest classes, *ES* and *OR* also show some chaotic variations, to a lesser extent. On the whole dataset, the AUC drops by 0.12 when the interfaces are shifted by 10%,corresponding to an F1-score of 0.9. A similar or even bigger gap is observed for all subsets comprising more than 15 proteins, except the enzyme-inhibitor subset (*EI*). On the whole dataset, the two antibody-antigen

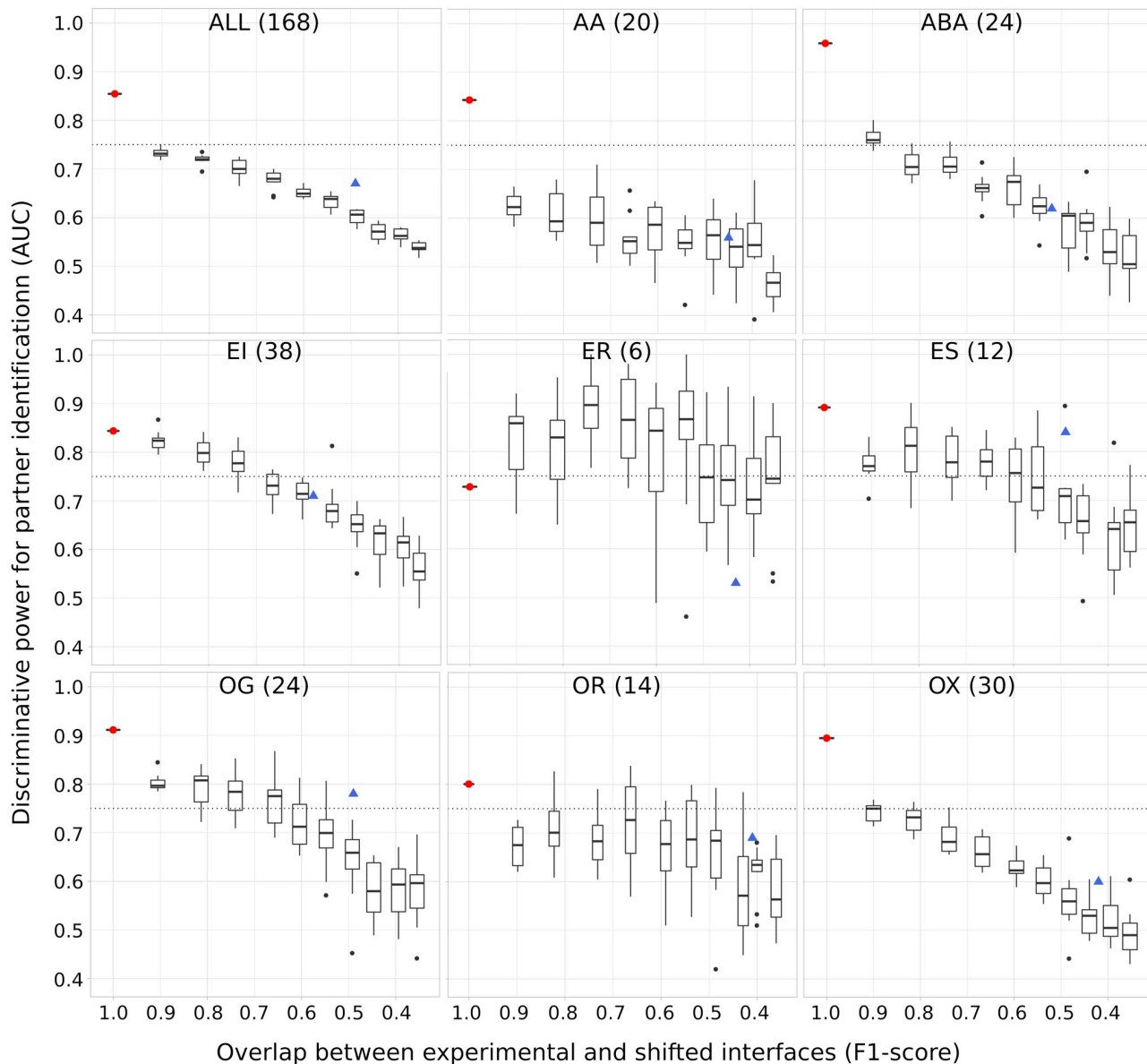

**Fig 4. Sensitivity of partner identification to approximations in the reference interfaces.** The RIs were obtained by gradually shifting the experimental interfaces (see Materials and methods). On each plot, we show 10 boxes corresponding to 10 different shift magnitudes. Each box comprises 10 AUC values obtained from 10 random generations of shifts in interfaces at a given amplitude. The values in x-axis give the average F1-scores computed for these shifted interfaces. The red dot and the blue triangle indicate the performance achieved using the experimental interfaces and the interfaces predicted by SC-dockSeed-mix as RIs, respectively. To compute the AUCs, we used the parameters identified as the best ones when using the experimental interfaces as RIs, namely a distance threshold of 6Å, the MAXDo docking energy, and without CIPS.

subsets (*AA* and *ABA*) and the *other* subset (*OX*), we identify cognate partners with en AUC lower than 75% with shifted decoys that still match very well (F1-score >0.8) the experimental interfaces. This shows that many competing proteins are able to bind favourably to almost the same protein surface region as the cognate partner. Compared to the shifted interfaces, our predicted interfaces allow reaching a similar or better partner discrimination for all classes but *ER*.

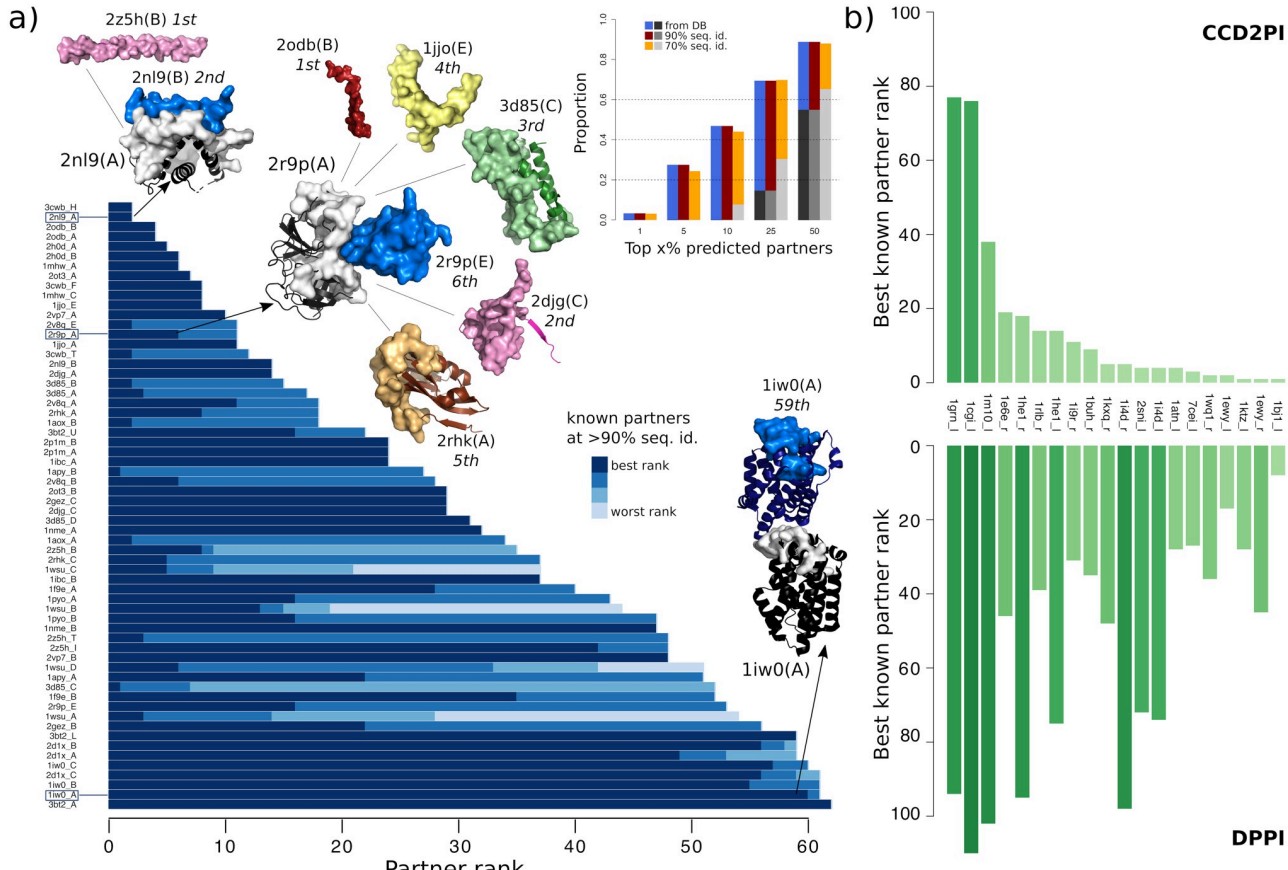

**Fig 5. Assessment of CCD2PI on an independent dataset, and comparison with a sequence-based deep learning method. (A)** The main barplot gives the rank(s) determined by CCD2PI for the known partner(s) of each protein from the independent dataset. The partners are inferred from the complex PDB structures involving the proteins from the set or their close homologs, sharing more than 90% sequence identity (see S10 Fig for the 70% sequence identity level). There are up to 4 partners for each protein, and they can be distinguished by the blue tones. The experimental structures of 3 cognate complexes are depicted as cartoons, with the query protein in dark grey and the best-ranked known partner in dark blue. For 2nl9:A and 2r9p: A, we also show, in other colors, the "incorrect" partners that obtained better ranks than the best-ranked known partner. For the complex made of two copies of 1iw0:A, the position and orientation of the copies was taken from the PDB structure 1wzg. The surfaces represent the RIs. The barplot in inset gives the proportion of proteins with at least one known partner in the top x% predictions. The grey bars give baseline values expected based on the number of known partners (see Materials and methods). **(B)** Comparison with DPPI. Best known partner ranks obtained from CCD2PI (on top) and DPPI (at the bottom). We focus on the subset of proteins for which the ranks provided by CC2PI are better (see S11 Fig for the full distributions).

## Accounting for protein surface multiple usage

Next, we assessed CCD2PI on an independent set of 62 proteins for which we defined some *interacting regions* accounting for the multiple usage of a protein surface by several partners and for molecular flexibility [23]. More precisely, we obtained each *interacting region* by merging overlapping interacting sites detected in the biological assemblies (from the PDB) involving the protein itself or a close homolog, as described in [23]. These regions can be seen as binding "platforms" for potentially very different partners. In this experiment, we used predicted interfaces as RIs, and all of them match well the experimentally known interacting regions (F1-score>0.6). CCD2PI identifies at least one known partner in the top 3 predictions (3/62 = 5%) for about a third of the proteins (Fig 5A, inset). For instance, CCD2PI identifies the Bcl-2-like protein 11 (2nl9:B), known partner of the Mcl-1 protein (2nl9:A), at the second position. It ranks first a tropomyosin construct (2z5h:B) that folds into an $\alpha$-helical shape similar to that of the known partner. For trypsin-3 (2r9p:A), five proteins are predicted as better

binders than its known inhibitor (2r9p:E). An extreme example is given by the heme oxyge-nase (1iw0:A), whose interaction with itself is very poorly ranked. This may be explained by the fact that the homodimer is asymmetrical, with two different interaction sites for the two copies, one of them not being taken into account by CCD2PI.

### Comparison with a sequence-based deep learning approach

Finally, we compared CCD2PI with DPPI [61], a deep learning method predicting protein interactions from sequence information only. DPPI takes as input two query proteins, each represented by a sequence profile, and outputs a score reflecting the probability that they phys-ically interact. The parameters of the architecture are learnt from experimentally known inter-actions. We re-trained the architecture to assess its performance on PPDBv2 (see Materials and methods). DPPI is able to single out the known partners (annotated in the database or inferred at >90% identity) with a very high accuracy, reaching an AUC of 95% versus 79% for CCD2PI. Yet, for a subset of 20 proteins, we obtained better ranks for the known partners (Fig 5B). These proteins belong to different functional classes. Two of them, namely 1i4d_r and 1he1_r (according to the PPDBv2 nomenclature) are copies of the human Rac GTPase (Uni-prot id: P63000). In total, Rac GTPase appears in three complexes from PPDBv2, 1i4d, 1he1 and 1e96, where it interacts with its three known partners. While the three partners are identi-fied in the top 5 by DPPI when using 1e96_l as the query, they are ranked between 95 and 101 when using 1i4d_r or 1he1_r.The three query sequences display near-perfect sequence identi-ties, but they cover more or less extended portions of the protein. Hence, the discrepancy between the results reveals a substantial sensitivity of DPPI with respect to different sequence contexts. The lack of a detection may be explained by an altered balance between signal and noise or between different signals coming from different interactions, or by some missing out-of-interface signal relevant for the interaction. In that case, we observed that our docking-based approach is more robust, as it finds at least one partner in the top 18 whatever the query.

## Discussion

We have addressed the issue of predicting protein-protein interaction networks from the per-spective of structural modelling, which is a useful complement to the machine learning sys-tems working with sequences and trained on experimental data.We have proposed a general approach to identify protein partners from large-scale docking experiments. We found that cognate partners can be singled out with high accuracy within specific functional classes. Beyond this parameter, we have identified a number of factors contributing to improving the discriminative power of the approach. We have primarily placed ourselves in a context where we seek to identify only one "true" partner for a given protein, while the other studied proteins are considered as non-interactors. We have found that in such conditions, the definition of the binding interface should be very precise to allow achieving high discriminative power. This requirement could be alleviated by putting more weights on the docking energy and pair potential contributions in the calculation of the interaction index. Nevertheless, we highlighted a few cases where "incorrect" reference interfaces might actually correspond to alternative binding modes.

   In reality, most proteins interact with multiple partners, via overlapping or distinct regions at their surface. Our current knowledge and understanding of the multiplicity of protein sur-face usage is still very limited. To move forward, we have collected experimentally character-ised protein complexes among the proteins in our benchmark set and also among their close homologs. The rationale was that protein interactions tend to be conserved among close homologs, as evidenced by the success of homology-based prediction of protein complex 3D

structures. This analysis revealed many possible interactions between the studied proteins, and showed that these interactions tend to populate regions in our predicted matrices displaying high interaction strengths. Hence, the propensities of interaction inferred from docking agree with the available structural data. As more complexes will be structurally characterised, we expect that the "experimental" interaction matrix will resemble more and more the predicted one, *i.e.* with many dark spots (high values).

A limitation of both experimental structural data and our computational framework is that they often cannot determine whether a protein-protein interaction will be functional or not in the cell. For instance, many antibody-antigen interactions can be inferred by homology transfer while the specificity of such interactions is very high and determined by only a few residues. A previous cross-docking study also highlighted the importance of the backbone conformation of the antibody to obtain a high-quality docked interface and thus be able to discriminate binders from non-binders [62]. More generally, the role of short peptide motifs for substrate selectivity and protein specific functions is being widely recognised [63], and there are documented examples of enzymes sharing high sequence identity while targeting different substrates [64]. Sequence-based learning approaches may overcome these limitations, but they do not provide direct information about the role of each residue in the formation and/or stabilisation of the assembly yet. By providing a 3D geometrical and physico-chemical description of the interactions at the residue level, our approach can help to reason about sequence-based predictions. For instance, we observed some common patterns shared between the proteins competing for the same partners. A systematic analysis of the effect of the sociability-based normalisation on different parts of the interface could give clues about the specificity determinants of molecular recognition. Reciprocally, sequence-based motif or specificity-determining site detection approaches could help to guide the docking toward boosting the accuracy of complex configuration prediction and to improve functional annotations of protein interactions. Such a combination of approaches may be particularly useful to distinguish multiple (potentially overlapping) interfaces.

Complete cross-docking calculations between hundreds of proteins remain computationally demanding. Nevertheless, they can be efficiently parallelized on grid-computing systems. Here, the docking calculations were distributed on the public World Community Grid (www. worldcommunitygrid.org). For a more convenient usage on a personal computer, the approach can be applied to discover and characterise interactions between proteins involved in a particular metabolic or signaling pathway. For instance, one could use it to explore the interactions between the 11 enzymes of the Calvin-Benson cycle and their inhibitors/activators [65].

Very recent studies indicate that the AlphaFold2 deep learning system [66–68], or a modified version trained specifically for multimeric inputs [69], outperforms all *ab initio* docking algorithms and template-based docking approaches. It predicts acceptable conformations for about two thirds of the tested dimers, and estimates prediction quality with a very small error rate. Moreover, its ability to predict a near-native conformation can be used to discriminate partners from non-interactors. These exciting results suggest that interfaces, conformations and interaction strengths can be directly obtained for a large number of complexes. AlphaFold2 predictions could be included in our approach in several ways. High-quality models could be used to predict reference interfaces, either directly, or by contributing to the residue-based scoring schemes we have defined. The prediction quality estimates could serve as pair-specific weights in the calculation of the interaction indexes. In addition, we could restrict the docking calculations to the subset of pairs for which AlphaFold2 produces low-quality predictions. For instance, AlphaFold2 struggles with some eukaryotic complexes, with antibody-antigen complexes, and with complexes displaying small interfaces [66, 69]. In such cases, the

information provided by the deep learning system is limited to an unreliable conformation. By contrast, we have shown that the conformational ensembles generated by docking, even between non-interacting pairs, are useful to guide the prediction of interfaces, to gain insight into protein sociability, and to discover alternative binding modes and new partners.

## Materials and methods

### Protein datasets

The first dataset is the Protein-Protein Docking Benchmark 2.0 (PPDBv2) [56] (https://zlab.umassmed.edu/benchmark/), which comprises 168 proteins forming 84 binary complexes. Each protein may be comprised of one or several chains, and is designated as receptor (r) or ligand (l). For most of the proteins, we used the unbound crystallographic structures for the docking calculations. The 12 notable exceptions are antibodies for which the unbound structure is unavailable and the bound structure was used instead. As there are also unbound antibodies present in the dataset, we can evaluate the impact of conformational changes on the results. The complexes of PPDBv2 are grouped in eight classes following [70]: antibody-antigen (AA, 20 proteins), bound antibody-antigen (ABA, 24), enzyme-inhibitor (EI, 38), enzyme with regulatory or accessory chain (ER, 6), enzyme-substrate (ES, 12), other-with-G-protein (OG, 24), other-with-receptor (OR, 14) and others (OX, 30). Note that for three cases, namely 1IR9, 1KXQ and 2HMI, there was an inversion in the original dataset between receptor and ligand, which we fixed here.

The second dataset is the P-262 benchmark introduced in [23]. It comprises 262 single protein chains for which single and multiple partners interactions are known in the PDB. We used bound conformations found in complex structures for the docking calculations. This dataset was extracted from a larger set of 2246 protein chains defined in the scope of the HCMD2 project (see http://www.ihes.fr/∼carbone/HCMDproject.htm). Based on the information recovered from the PDB, the proteins were manually classified in eleven groups, following and extending the classification proposed [70]. Hence, the set is comprised of 16 bound antibodies (AB), 25 complex subunits (C), 60 enzymes (E), 10 enzyme regulators (ER), 9 G proteins (G), 6 antigens from the immune system (I), 23 receptors (R), 24 structural proteins (S), 16 substrates/inhibitors (SI), 7 transcription factors (TF) and 66 proteins with other function (O).

### Interacting pair identification by homology transfer

We extended the set of known partners by transferring knowledge from close homologs. Specifically, we exploited the pre-computed PDB homology clusters with 90% and 70% sequence identities. For each protein pair considered, we verified the existence of a physical contact between the proteins in the pair, or some homologs at 90% (resp. 70%) sequence identity. Two proteins were considered to be in a contact if their interface was larger than 5 residues, as detected by INTBuilder [60] (http://www.lcqb.upmc.fr/INTBuilder/). This procedure was performed at the protein chain level. To deal with the multi-chain proteins from PPDBv2, we considered that two proteins were in interaction whenever at least one pair of chains from the two proteins was in interaction.

### Cross-docking calculations

Given an ensemble of proteins, complete cross-docking consists in docking each protein against all the proteins in the dataset, including itself. All calculations were performed by the MAXDo (Molecular Association via Cross Docking) algorithm [54].

**Reduced protein representation.** The protein is represented using a coarse-grain protein model [42] where each amino acid is represented by one pseudoatom located at the C$\alpha$ position and either one or two pseudoatoms representing the side-chain (with the exception of Gly). Interactions between the pseudoatoms are treated using a soft Lennard Jones (LJ) type potential with parameters adjusted for each type of side-chain (see Table 1 in [42]). In the case of charged side-chains, electrostatic interactions between net point charges located on the second side-chain pseudoatom were calculated by using a distance-dependent dielectric constant $\epsilon = 15r$, leading to the following equation for the interaction energy of the pseudoatom pair $i, j$ at distance $r_{ij}$:

$$E_{ij} = \left(\frac{B_{ij}}{r_{ij}^8} - \frac{C_{ij}}{r_{ij}^6}\right) + \frac{q_i q_j}{15 r_{ij}^2} \tag{3}$$

where $B_{ij}$ and $C_{ij}$ are the repulsive and attractive LJ-type parameters respectively, and $q_i$ and $q_j$ are the charges of the pseudoatoms $i$ and $j$. More details about the representation can be found in [54].

**Systematic docking simulations.** MAXDo implements a multiple energy minimization scheme similar to that of ATTRACT [42] where proteins are considered as rigid bodies. For each protein pair, one protein (called the receptor) is fixed in space, while the second (called the ligand) is placed at multiple positions on the surface of the receptor. For each pair of receptor/ligand starting positions, different starting orientations are generated by applying rotations of the gamma Euler angle defined with the axis connecting the centers of mass of the 2 proteins. We used two different protocols to explore the docking space for our two datasets. In the case of PPDBv2, the whole surface of the receptor was probed by the ligand. This was guaranteed by generating starting positions that covered the whole surface and restraining the ligand motions during the simulation so as to maintain its center of mass on a vector passing through the center of mass of the receptor protein. As a result, the receptor and the ligand are treated differently and given en protein pair $P_1 P_2$, docking $P_1$ against $P_2$ is not equivalent to docking $P_2$ against $P_1$. More details about this protocol can be found in [53, 54]. In the case of P-262, the ensemble of starting positions was restricted using predictions from the JET method [13]. This reduced the docking search space by up to 50%. Moreover, the restrain was removed, so that the ligand was free to migrate to a position completely different from its starting position. Thus, for each couple of proteins $P_1 P_2$, considering $P_1$ as the receptor and $P_2$ as the ligand is essentially equivalent to the reverse situation where $P_2$ is the receptor and $P_1$ is the ligand. More details about this protocol can be found in [71].

**Computational implementation.** For each pair, several hundreds of thousands of energy minimizations were performed. As each minimization takes 5 to 15 s on a single 2 GHz processor, a CC-D of several hundreds of proteins would require several thousand years of computation. However, the minimizations are independent from each other and thus can be efficiently parallelized on grid-computing systems. Our calculations have been carried out using the public World Community Grid (WCG, www.worldcommunitygrid.org), with the help of thousands of internautes donating their computer time to the project. It took approximately seven months to perform CC-D calculations on the PPDBv2, and three years on the complete HCMD2 dataset (2246 proteins) from which P-262 is extracted. More technical details regarding the execution of the program on WCG can be found in [72]. The data analysis was partly realized on Grid'5000 (https://www.grid5000.fr).

## Data analysis

**Detection and prediction of interface residues.** The docked interfaces are defined by the sets of residues from the two partners closer than $d$ Å. They were computed using INTBuilder [60] (http://www.lcqb.upmc.fr/INTBuilder/), and we considered three values for $d$, 4.5, 5 and 6. The experimental interfaces were detected in the X-ray structures of the cognate complexes using the same tool and a distance $d$ of 5 Å.

The reference interfaces were predicted using a modified version of dynJET$^2$ [23] (http://www.lcqb.upmc.fr/dynJET2/), a software tool predicting interacting patches based on four residue descriptors. Specifically, dynJET$^2$ relies on three sequence- and structure-based properties of single proteins, *i.e.* evolutionary conservation, physico-chemical properties and local geometry (measured by the circular variance), and on a systemic property reflecting docking-inferred binding propensities (S4 Fig, see also [23] for more detailed definitions). dynJET$^2$ algorithm first detects the *seed* of the patch, then *extends* it and finally add an *outer layer* [12]. At each step, surface residues are selected using a combination of the four descriptors. Four scoring strategies are implemented, to cover a wide range of interfaces. The first one, $SC_{cons}$ detects highly conserved residues and then grows the patches with residues less and less conserved and more and more protruding, and likely to be found at interfaces based on their physico-chemical properties. The second one, $SC_{notLig}$ is a variant of $SC_{cons}$ where local geometry is accounted for in the seed detection step to avoid buried ligand-binding pockets. The third one, $SC_{geom}$ disregards evolutionary conservation and looks for protruding residues with good physico-chemical properties. The fourth one, $SC_{dock}$, defines patches exclusively comprised of residues frequently targeted in docking calculations. We refer to this group of *SCs* as *SC-juxt*. We modified dynJET$^2$ to create 9 additional scoring schemes grouped in 3 main strategies, namely *SC-mix*, *SC-monoSeed-mix* and *SC-dockSeed-mix* (S4 Fig). All 9 scoring schemes are variants of $SC_{cons}$, $SC_{notLig}$ and $SC_{geom}$ including the docking-inferred binding propensities in different ways. *SC-mix* combines them with the other descriptors at each step. *SC-monoSeed-mix* detects the seeds using only the single-protein based properties, and then combines the latter with the docking propensities to grow the patches. *SC-dockSeed-mix* relies exclusively on the docking propensities to detect the seeds and then grows them using a combination of all four descriptors. We implemented all scoring schemes in dynJET$^2$. For each protein, given a chosen main strategy, we detected a set of predicted patches using all its scoring schemes. Each patch was defined as a consensus of at least 2 iterations over 10 of dynJET$^2$. We then retained the patch or combination of patches matching the best the experimentally known interfaces.

We also used shifted decoys as reference interfaces. To generate them, we gradually shifted the experimentally known interfaces from the PPDBv2. For each experimental interface, we randomly generated 100 decoys, by moving between 10% and 100% of its residues. More precisely, the first 10 decoys were generated by moving 10% of the residues, the next 10 by moving 20%, etc. . . At each step of the algorithm, we randomly pick up an interface residue $r_s$ located at the border, *i.e.* at less than 5 Å of a surface residue that is not part of the interface. Then, we identify the interface residue located the farthest away from $r_s$, and we randomly pick up one of its neighbours $r_n$ ($< 5$ Å). We then switch the status of $r_s$ and $r_n$. In other words, $r_s$ is removed from the interface and $r_n$ is added to the interface. The residue $r_s$ cannot be picked again in the following iteration.

**Re-scoring of the docking models.** We considered three scoring functions, namely iAT-TRACT [73], PISA [74] and CIPS [59], in replacement or complement of the one implemented in MAXDo.

iATTRACT [73] is a docking software more recent than MAXDo and mixing a rigid-body docking approach with flexibility. The energy function is similar to that of MAXDo, except

that the repulsive term in the Lennard-Jones potential decreases more rapidly with the inter-atomic distance while the electrostatic contribution decreases less rapidly. Specifically, iAT-TRACT interaction energy of the pseudoatom pair $i$, $j$ at distance $r_{ij}$ is expressed as

$$E_{ij} = \left(\frac{\sigma_{ij}}{r_{ij}}\right)^{12} - \left(\frac{\sigma_{ij}}{r_{ij}}\right)^6 + \frac{q_i q_j}{\epsilon r_{ij}} \tag{4}$$

where $\sigma_{ij}$ is the LJ-type parameter, $q_i$ and $q_j$ are the charges of the pseudoatoms $i$ and $j$, and the dielectric constant $\epsilon$ is set to 10. Each of the docking models obtained from the CC-D was subjected to iATTRACT's minimisation process and we used the energy value coming from this minimization.

PISA [74] is a scoring method developed to discriminate between biological and non biological complexes. It relies on the dissociation free energy to evaluate the stability of a complex. On top of the dissociation free energy, PISA considers larger assemblies more probable than the smaller ones and considers that single-assembly sets take preference over multi-assembly sets. We used PISA to re-score the docking conformations produced by MAXDo.

CIPS [59] (http://www.lcqb.upmc.fr/CIPS/) is a statistical pair potential meant to be used as a high throughput technique able to largely filter out most of the non-native conformations with a low error rate. It was trained using 230 bound structures from the Protein-Protein Docking Benchmark 5.0 [75]. We used it to obtain complementary scores on the docking conformations.

**The protein interaction index—II.** We evaluate docking models using an interaction index $II$ computed as a product between three terms (see Eq 1). For a given protein pair $P_1 P_2$, the first term, $FIR_{P_1,P_2}$, is the overall fraction of the docked interfaces composed of residues belonging to the reference interfaces for the two proteins: $FIR_{P_1,P_2} = FIR_{P_1} * FIR_{P_2}$. It reflects the agreement between the docked interfaces and the reference interfaces. The reference interfaces may be experimentally known or predicted. The second one, $E_{P_1,P_2}$, is the docking energy provided by MAXDo, PISA or iATTRACT. The third one, $PP_{P_1,P_2}$ is the value computed by CIPS and it may or may not be included in the formula. The product is computed for every docking conformations and the minimum (best) value is kept.

**The protein normalized interaction index—NII.** To account for the global social behavior of the proteins, we further normalize the interaction indices. The normalized interaction index $NII$ between $P_1$ and $P_2$ was determined as

$$NII_{P_1,P_2} = \frac{min(II'_{P_1,P_2}, II'_{P_2,P_1})^4}{min_P(II'_{P_1,P}) \cdot min_P(II'_{P,P_2}) \cdot min_P(II'_{P,P_1}) \cdot min_P(II'_{P_2,P})} \tag{5}$$

where $II'_{P_1,P_2}$ is a symetrized weighted version of the interaction index $II_{P_1,P_2}$ and it is defined as:

$$II'_{P_1,P_2} := \frac{II_{P_1,P_2}}{\sqrt{S_{P_1} \cdot S_{P_2}}}, \, S_{P_i} := \frac{1}{2|\mathcal{P}|} \sum_{P_j \in \mathcal{P}} II_{P_i,P_j} + II_{P_j,P_i} \tag{6}$$

where $\mathcal{P}$ is the ensemble of proteins considered. The normalization can be applied to the whole dataset or to subsets. In either case, $NII$ values vary between 0 and 1. For each protein $P_i$, we defined its predicted partner as the protein $P_j$ leading to $NII_{P_i,P_j} = 1$.

**Parameter setting.** The four main parameters of our approach and the different values we considered are reported in Table 1. They were optimized on the PPDBv2. For each subet, we computed 72 AUC values corresponding to the 72 possible combinations of parameter values. Then, we ranked the combinations based on their weighted average AUC values. Given a

combination $C_i$, the average was computed as

$$\overline{AUC}(C_i) = \frac{\sum_{j=1}^{n}(N_j \times AUC^j(C_i))}{\sum_{j=1}^{n} N_j}, \qquad (7)$$

where $N_j$ is the number of proteins in the subset $j$ and $n$ is the number of subsets. We considered as subsets the eight functional classes and also the entire dataset itself, leading to $n = 9$. The weighting minimises the effect a subset with a low number of proteins could have on the global ranking, while putting more importance on subsets with a large number of proteins. The combination maximizing the value of $\overline{AUC}(C_i)$ was chosen as the default one (Table 1, in bold).

Then, for each class $j$, we ranked the 72 possible combinations according to their AUC values, $AUC^j(C_i)$, and we retained the top 20%, hence 15 combinations. This pool was separated by each one of the four parameters. Whenever we found a parameter value leading to a better AUC than the default value, we further assessed this difference with a Mann Whitney U-test [76, 77]. For this test, we went back to the whole ensemble of 72 combinations and compared the distributions of AUC values obtained with the default value and the other value, respectively. If the p-value was lower 0.01, then we considered the other value to significantly improve our discrimination potency over the default one. And we decided to use it for the given class.

We applied the same procedure when dealing with the experimental interfaces. Since the number of possible combinations (18) is much lower in that case, we retained the top 30%, hence 6 combinations.

**Assessment of the predictions.** We compute the proportion of proteins with at least one known partner in the top $X$% predictions as:

$$P_{topX} = \frac{1}{N}\sum_{i=1}^{N} \mathbb{1}_{\min_k(R_k^i) \leq \frac{X}{100}N}, \qquad (8)$$

where $R_k^i$ is the rank of the kth partner predicted by CCD2PI for protein $P_i$, and $N$ is the total number of proteins in the set. We put this proportion in context with respect to some baseline value computed by counting the number of times we expect to find at least one known partner in a randomly chosen subset:

$$P_{topX}^{base} = \frac{1}{N}\sum_{i=1}^{N} \mathbb{1}_{\frac{X}{100}N_i \geq 1}, \qquad (9)$$

where $N_i$ is the number of known partners for protein $P_i$.

## Comparison with DPPI

We re-trained DPPI architecture [61] on the Profppikernel database [78] containing 44 000 interactions (10% positive). The positive samples were taken from the HIPPIE database [79]. We removed from the training set all sequences which share more than 70% identity with any sequence from PPDBv2. We clustered the samples such that any two sequences do not share more than 40% identity. We used MMseqs2 [80] to cluster and filter sequences.

## Supporting information

**S1 Fig. Number of putative partners predicted by CCD2PI.** Each grey curve corresponds to a protein from the PPDBv2, and indicates the number of putative partners (y-value) with a *NII*

greater than a threshold (x-value). The red curve shows the average behaviour.
(TIF)

**S2 Fig. Predicted interaction matrices for the PPDBv2. (A-B)** Matrices computed using predicted interfaces as references. **(C-D)** Matrices computed using experimental interfaces as references. The matrices on the left give interaction indices (*II*) and those on the right the normalized interaction indices (*NII*).
(TIF)

**S3 Fig. Comparison of cognate partners and competitors interfaces.** The "interaction strength" is used for plotting pairs with respect to *II* values (grey) and *NII* values (blue). The 168 proteins are ordered along the x-axis according to the *II* ranks they obtained with the protein of interest, and for each position on the x-axis, two points are plotted. The point corresponding to the *NII* value of the cognate partner is highlighted in red. The *II* values are scaled between 0 and 1. The predicted RIs for the cognate partner and the top 5 competitors are depicted as surfaces colored by amino acid properties: positive (KR) in blue, negative (DE) in red, polar (HNQST) in cyan, aromatic (FWY) in pink, hydrophobic (AGILMV) in white, cysteine (C) in yellow, and proline (P) in green. The boxplots show the distribution of the proportion of positives (panel a) or negatives and aromatic (panel b) residues in the RIs. The values for the cognate partner and the top 5 competitors are indicated by colored dots.
(TIF)

**S4 Fig. Properties of the known interacting pairs. (A-B)** Distributions of the number of partners, for each protein within each subset, inferred by homology at 90% (a) and 70% (b) sequence identity levels. **(C)** Cross-interaction density, defined as the percentage of cells corresponding to a known interaction, within the matrix associated to each subset. The two grey tones indicate the sequence identity level. **(D)** Agreement between cross-interaction density and predicted NII values. In x-axis are reported the ratios $r_k = \max\left(\frac{\sum_{i,j \in S_k} NII_{R_i,R_j}}{\sum_{i,j \in S_k} NII_{L_i,L_j}}, \frac{\sum_{i,j \in S_k} NII_{L_i,L_j}}{\sum_{i,j \in S_k} NII_{R_i,R_j}}\right)$.

For each subset $S_k$, $r_k$ reflects the difference in predicted interaction strengths among the receptors versus the ligands. When the dot is grey, it means the receptors are predicted to interact more with each other, while a red dot indicates that the ligands interact more. In y-axis are reported the different of cross-interactions densities between receptors and ligands, or reciprocally. When the value is positive, it means the tendency observed for the known interactions agrees with that observed for the predictions. For instance, antibodies are predicted to interact with each other twice more than antigens, and there are 50% more known interactions between them. Known interactions were determined with a sequence identity level of 70%.
(TIF)

**S5 Fig. Examples of annotated and homology-transferred interactions.** The query protein is represented as a grey cartoon. The cognate partner annotated in the PPDBv2 is shown in blue and a partner identified in the PDB by homology transfer (>90% sequence identity) is shown in dark red. In each case, the proteins come from the same functional class: **(A-B)** other-with-G protein, *OG*, **(C)** others, *OX*. The intra-class ranks of the partners are given.
(TIF)

**S6 Fig. Scoring schemes used to predict interfaces.** Each scoring scheme is depicted by a schematized representation of a predicted patch, where the different concentric layers correspond to different combinations of four residue-based descriptors. $T_{JET}$: evolutionary conservation. *PC*: physico-chemical properties. *CV*: circular variance. *NIP*: docking-inferred binding propensities. **Top left panel:** *SC-juxt* comprises four scoring schemes, three of them

($SC_{cons}$, $SC_{notLig}$ and $SC_{geom}$) using $T_{JET}$, $PC$ and $CV$ and the fourth one ($SC_{NIP}$) exclusively based on $NIP$. $SC_{cons}$ detects highly conserved seeds and extend them using physico-chemical properties and local geometry. $SC_{notLig}$ is a variant of $SC_{cons}$ including circular variance at the seed detection step to avoid buried ligand-binding pockets. $SC_{geom}$ disregards evolutionary conservation and detects protruding regions with good phyisco-chemical properties. All other scoring schemes are variants of $SC_{cons}$, $SC_{notLig}$ and $SC_{geom}$ including $NIP$ in different ways. **Top right panel:** *SC-mix* combines $NIP$ with the other descriptors at each step. **Bottom left panel:** *SC-monoSeed-mix* disregards $NIP$ to detect the seeds and then combines it with the other descriptors. **Bottom right panel:** *SC-dockSeed-mix* relies exclusively on $NIP$ to detect seeds and then uses a combination of the four descriptors.
(TIF)

**S7 Fig. Detailed predictive performance for PPDBv2, when using the knowledge of the experimental interfaces. (A)** Comparison of the AUC values obtained when the parameters were optimized for dealing with experimental interfaces or for dealing with predicted interfaces. The parameters for experimental interfaces are a 6 Å threshold, the MAXDo energy function and no CIPS. They were applied to all classes but EI, where PISA was used instead of MAXDo. The parameters for predicted interfaces are a 5 Å threshold, the MAXDo energy function and CIPS. There are three exceptions: PISA was used for EI, iATTRACT was used for ER and CIPS was not used for OR. **(B-D)** Influence of the individual parameters on the predictive performance. **(B)** Distance threshold used to define docked interfaces. **(C)** Docking energy. **(D)** Presence or absence of the CIPS pair potential, depending of the docking energy. In each plot, for each protein class, we considered the 6 combinations with the highest AUC values. This pool of combinations was divided into 2 to 4 subsets depending on the number of values considered for the parameter. The opaque bars indicate the average AUC values computed over the subsets of combinations. The parts in transparent indicate the maximum values. If a parameter value was not present in the 6 best combinations, then it does not appear on the plot.
(TIF)

**S8 Fig. Influence of the quality of the interface predictions on partner identification.** The AUC values are plotted in function of the F1-score and the positive predictive value (PPV) of the predicted RIs, for the whole dataset and a subset of classes (each containing more than 15 proteins). On each plot, the red line corresponds to a linear regression between the two variables, whose adjusted $R^2$ is reported in the top left corner. The scoring strategy is SC-dockSeed-mix and the AUC values correspond to CCD2PI default parameter setting.
(TIF)

**S9 Fig. Partner identification for proteins where the predicted and experimental interfaces do not match (F1-score < 0.2).** For each protein, we show the improvement (in green) or the deterioration (in red) of the native partner's rank upon replacing the predicted RIs with the experimental interfaces. The ranks obtained using the predicted RIs are marked with horizontal ticks—the other extremity of the segment corresponding to using the experimental RIs. The partner is identified either within the whole PPDBv2 (left segment) or only within the functional class of the protein (right segment).
(TIF)

**S10 Fig. Assessment of CCD2PI on an independent dataset.** For each protein from the set, the barplot indicates the rank(s) determined by CCD2PI for its known partner(s). The partners are inferred from the complex PDB structures involving the proteins from the set or their close homologs, sharing more than 70% sequence identity. There are up to 12 partners for each

protein, and they can be distinguished by the blue and purples tones. Compare with Fig 5A in the main text.
(TIF)

**S11 Fig. Comparison with DPPI on PPDBv2.** Distributions of the best ranks predicted by DPPI (left, lightblue) and CCD2PI (right, lightgreen) for the known partners, inferred at 90 and 70% sequence identity levels.
(TIF)

## Acknowledgments

We thank the World Community Grid (WCG, www.worldcommunitygrid.org) and its volunteers for allowing us to perform cross-docking experiments with MAXDo on the PPDBv2.0.

## Author Contributions

**Conceptualization:** Elodie Laine, Alessandra Carbone.

**Data curation:** Chloé Dequeker, Laurent David, Elodie Laine, Alessandra Carbone.

**Formal analysis:** Chloé Dequeker, Yasser Mohseni Behbahani, Elodie Laine, Alessandra Carbone.

**Funding acquisition:** Alessandra Carbone.

**Investigation:** Elodie Laine, Alessandra Carbone.

**Methodology:** Elodie Laine, Alessandra Carbone.

**Software:** Chloé Dequeker, Yasser Mohseni Behbahani.

**Supervision:** Elodie Laine, Alessandra Carbone.

**Validation:** Elodie Laine, Alessandra Carbone.

**Visualization:** Elodie Laine.

**Writing – original draft:** Elodie Laine, Alessandra Carbone.

**Writing – review & editing:** Elodie Laine, Alessandra Carbone.

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
