## [Decision Letter · Decision Letter 0]

14 Oct 2021

Dear Prof. Carbone,

Thank you very much for submitting your manuscript "From complete cross-docking to partners identification and binding sites predictions" for consideration at PLOS Computational Biology.

As with all papers reviewed by the journal, your manuscript was reviewed by members of the editorial board and by several independent reviewers. In light of the reviews (below this email), we would like to invite the resubmission of a significantly-revised version that takes into account the reviewers' comments.

We cannot make any decision about publication until we have seen the revised manuscript and your response to the reviewers' comments. Your revised manuscript is also likely to be sent to reviewers for further evaluation.

Sincerely,

Rachel Kolodny

Associate Editor

PLOS Computational Biology

Nir Ben-Tal

Deputy Editor

PLOS Computational Biology

Reviewer's Responses to Questions

**Comments to the Authors:**

Reviewer #1: Dequeker et al. present a protein-protein cross docking approach to identify binding partners for a given set of proteins. It is based on previous work by the Carbone group and indicates significant improvements. Prediction of potential binding partners of proteins is one major goal of bioinformatics and computational biology. The route taken by the authors is interesting and differs from other approaches (mostly based on machine learning trained by experimental input data). The prediction is based on cross docking of all against all proteins in the data set (structures of the proteins need to be known). The approach has the potential to identify protein-protein interaction pairs that cannot be detected based on homology to some known interacting pair. I have, however, some comments:

1. Although the approach is quite clearly explained the scoring and selection of correct complexes is sometimes difficult to understand. As a scoring function the authors use a product of overlap of the interface of a docked structure and the Energy of the docked complex and the pair potential energy. It is not clear why the authors use here a product (as interaction index) and not for example the sum of these contributions (appropriatly weighted).

2. In the scoring the authors use a predicted "reference interface" (not the real native interface) for comparison with a docking interface. This type of reference interface may differ from the correct "native interface". Did the authors check how often this is the case? In such case an incorrect structure would be favored by the score!

3. The authors identify the "correct" partner for a protein with reasonable accuracy (AUC 0.67) and claim that there are only few predicted alternative high affinity (incorrect) complexes. However, my impression by looking at Fig. 2 is that there are quite a number of "dark" spots in the interaction matrix such that there are something like 10-20 putative PPIs for each given protein (on average)?

4. I think it could be useful to look in more detail at some of the "incorrect" predictions. What is the main reason that some incorrect predicted complexes get high interaction scores and hence would be predicted as forming a realistic complex? This is not dicussed.

5. The approach is very computationally demanding which could be a major drawback in putaitve applications. The authors should extend the discussion section to indicate possible sets of proteins (containing a limited number of partners) for which an application of the approach could be useful.

Reviewer #2: Prediction of protein-protein interaction networks is one of the most important biological problems. Experimental techniques for determination of such networks, based on high-throughput methodologies are known to be not very accurate. Computational approaches based on sequence information have been around for a long time, and also suffer from limited accuracy. This study continues the authors' series of publications addressing this issue from the perspective of structural modeling, which is a useful complement to the experimental and sequence-based techniques. In this study, the authors added more structural and physicochemical parameters to their previously used set. The new, expanded set showed a significantly better discrimination power to the non-cognate interactors, thus considerably improving the utility of the method. The paper contains detailed analysis of the approach, parameters variation, and performance on the general and function-specific protein-protein sets, as well as comparison with alternative approaches. The approach is a useful addition to the arsenal of computational techniques for characterization of protein interactions, and as such would be of interest to the biological community.

Reviewer #3: The authors propose a molecular docking approach to predict protein partners and interaction strength. They assessed their algorithm for recovery of known and predicted protein-protein interactions, finding that their algorithm improves upon previous partner identification methods, although it is outperformed by a deep learning method.

MAJOR CONCERNS

The pair extension by homology is fundamental to the paper, but the authors do not provide sufficient evidence for its validity, or for the thresholds used. The authors acknowledge its limitation, but do not provide sufficient explanation or references. “We considered that a structurally characterized interaction found for P1′ and P2′, homologs of P1 and P2, respectively, was a strong indicator of the possibility for P1 and P2 to interact with each other. Nevertheless, we should stress that homology transfer does not guarantee that the interaction between P1 and P2 is functional in the cell.“

The docking-based partner prediction approach shown here is significantly outperformed by existing machine learning methods (AUC 95% versus 79% recovery of known and inferred partners). The authors are not convincing in the need for their method given the poor performance compared to existing methods.

Authors use different levels of homology to define true pairs between different datasets and figures (Figure 2, 5).

The probability at random (grey bars) in figure 2b is unclear.

Figure 5a inset is unclear, what are the grey bars? Additionally, this figure does not clearly support “CD2PI identifies at least one known partner in the top 3 for about a third of the 319 proteins (Fig. 5a, inset)”

Figure 5a is unclear. What are the structures shown in colors? Caption describes these structures as known partners, but there are not 6 blue tones for 2r9p:A.

Figure 5b should include all proteins, not just cases where CCD2PI outperforms DPPI

MINOR POINT

“We combine together physics-based energy, interface matching and protein sociability, three ingredients we previously showed to be relevant to partner identification and discrimination.” Lacks citation.

In summary, some figures in the paper are somewhat difficult to interpret. The evidence supporting pair extension by homology is moderate, and the approach is significantly outperformed by existing methods.

**Have the authors made all data and (if applicable) computational code underlying the findings in their manuscript fully available?**

Reviewer #1: Yes

Reviewer #2: Yes

Reviewer #3: Yes

PLOS authors have the option to publish the peer review history of their article (what does this mean?). If published, this will include your full peer review and any attached files.

Reviewer #1: **Yes: **Martin Zacharias

Reviewer #2: **Yes: **Ilya Vakser

Reviewer #3: No
---

## [Decision Letter · Decision Letter 1]

3 Jan 2022

Dear Prof. Carbone,

Thank you very much for submitting your manuscript "From complete cross-docking to partners identification and binding sites predictions" for consideration at PLOS Computational Biology. As with all papers reviewed by the journal, your manuscript was reviewed by members of the editorial board and by several independent reviewers. The reviewers appreciated the attention to an important topic. Based on the reviews, we are likely to accept this manuscript for publication, providing that you modify the manuscript according to the review recommendations.

Sincerely,

Rachel Kolodny

Associate Editor

PLOS Computational Biology

Nir Ben-Tal

Deputy Editor

PLOS Computational Biology

[LINK]

Reviewer's Responses to Questions

**Comments to the Authors:**

Reviewer #1: I carefully read the new version of the manuscript and the responses to my concerns. I think the authors successfully responded to my concerns. The manuscript is an interesting approach to predict new putative protein-protein interactions. Very recently, AI approaches such as Alphafold2 have become very successful in predicting not only protein structures but also putative protein-protein interactions. I think it could be very valuable for the manuscript to relate/discuss the "docking" approach of the authors vs. emerging AI based approaches to predict complex structure. This could further improve the manuscript.

**Have the authors made all data and (if applicable) computational code underlying the findings in their manuscript fully available?**

Reviewer #1: Yes

PLOS authors have the option to publish the peer review history of their article (what does this mean?). If published, this will include your full peer review and any attached files.

Reviewer #1: No

Figure Files:

Data Requirements:

Reproducibility:

References:

---

## [Editor Report · Decision Letter 2]

11 Jan 2022

Dear Prof. Carbone,

We are pleased to inform you that your manuscript 'From complete cross-docking to partners identification and binding sites predictions' has been provisionally accepted for publication in PLOS Computational Biology.

Best regards,

Rachel Kolodny

Associate Editor

PLOS Computational Biology

Nir Ben-Tal

Deputy Editor

PLOS Computational Biology

---

## [Editor Report · Acceptance letter]

24 Jan 2022

PCOMPBIOL-D-21-01580R2 

From complete cross-docking to partners identification and binding sites predictions

Dear Dr Carbone,

I am pleased to inform you that your manuscript has been formally accepted for publication in PLOS Computational Biology. Your manuscript is now with our production department and you will be notified of the publication date in due course.

With kind regards,

Anita Estes
